# Establishment of *Echinococcus granulosus EgM123* Recombinant Gene Rabies Virus SRV_9_ and Identification of Its Biological Characteristics

**DOI:** 10.3390/v17010030

**Published:** 2024-12-28

**Authors:** Yueqi Yang, Mengdan Hou, Guicheng Su, Xiaoyan Ma, Xiaohui Su, Kunlei Li, Songhan Liu, Luheng Xiao, Jingjing Yao, Jiahao Zhai, Xiaoying Wei, Yang Zhou, Qianqian Lai, Yuwei Dong, Jieyu Liu, Shaohua Zhai

**Affiliations:** 1School of Veterinary Medicine, Xinjiang Agricultural University, Urumqi 830052, China; yangyueqi7788@163.com (Y.Y.); 15770022711@139.com (M.H.); ebh089@outlook.com (K.L.); a807953546@2980.com (S.L.); xiaoluheng0107@163.com (L.X.); 18703021672@163.com (J.Y.); 18195862757@163.com (J.Z.); weixxiy2023@163.com (X.W.); 18324021810@163.com (Y.Z.); lqq369948@163.com (Q.L.); dyw030616vvv@163.com (Y.D.); 15760102749@163.com (J.L.); 2Livestock and Veterinary Work Station of Xinjiang Production and Construction Corps, Urumqi 830063, China; 13999831217@163.com (G.S.); mxy0313@163.com (X.M.); sxh19890125@sina.com (X.S.)

**Keywords:** rabies virus SRV_9_, *Echinococcus granulosus EgM123* gene, reverse genetics, virus rescue

## Abstract

Canids act as a crucial intermediary in the transmission of rabies and *Echinococcus granulosus*, serving as co-infection hosts and pathogen carriers for both rabies and hydatid disease (HD) transmitted from animals to humans. Therefore, an effective and efficient bivalent oral vaccine for preventing HD and rabies is urgently required to reduce economic losses in husbandry resulting from rabies and HD. In this study, a full-length plasmid (pcDNA4-NPM+G_ΔCD_+EgM123+eGFP+L) carrying the *Echinococcus granulosus EgM123* gene and fluorescence reporter genes of eGFP and four auxiliary transfection plasmids of rabies virus SRV_9_ (pcDNA4-N, pcDNA4-P, pcDNA4-G, pcDNA-L) were established by reverse genetics approaches and co-transfected to BSR cells by electrotransfection. The co-transfected BSR cells showed green fluorescence 48 h after electrotransfection. The recombinant virus was exposed to the sixth-generation blind passage, with the *N*, *P*, *G*, and *EgM123* genes amplified via RT-PCR, yielding targeted strips. The rescued virus-infected BSR cells were characterized by TEM, and the results indicated that bullet-like viral particles with an average size of 148.47 nm and a cyst structure were present in the cytoplasm of BSR cells; the expression levels of continuously cultivated 9th-, 10th-, 11th-, 12th-, and 13th-generation viruses were quantified by qRT-PCR, and the results showed that mRNA expression of the virus was upregulated. The LD_50_ titer of suckling rats was measured to be 10^−1.4^. The synthesized *EgM123* recombinant gene rabies virus SRV_9_ can function as a vaccine strain for the development of the “Rabies-HD bivalent recombinant gene oral vaccine”, therefore aiding in the prevention and management of rabies and HD in animals.

## 1. Introduction

Rabies, also known as hydrophobia, is an acute, lethal, and paralytic zoonotic infection caused by the rabies virus (RV) of the Lyssavirus in the Rhabdoviruses family [1]. The RV is mainly found in the salivary glands and saliva of affected animals. When an animal infected with rabies attacks people or other animals, the virus can enter the victim’s body via saliva, resulting in infection [2]. The invasion of the virus into the central nervous system typically results in irreversible neurological damage to the host, ultimately leading to the host’s death, with a fatality rate of 100% [3]. The World Health Organization (WHO) proposed a global strategic plan in 2018 to eradicate human rabies deaths by 2030 [4]. Currently, the main prevention and control difficulty of rabies lies in immunizing wildlife in developed countries and stray dogs in developing countries with rabies vaccine [5]. Therefore, oral vaccination against rabies for stray dogs and wild animals is necessary and the most efficacious strategy for preventing and controlling rabies [6].

Rabies virus SRV_9_, a low-virulent strain used for the study of oral vaccines, is cloned from the parental strain SAD B19, whose genome-specific locus mutation results in weakened virulence and is suitable for the study of an attenuated oral vaccine [7]. Wang et al. [8] constructed a full-length cDNA plasmid of SRV_9_ based on the vaccine of rabies virus SRV_9_ using eukaryotic expression vector pCI and pcDNA3.1(+). A reporter gene was inserted between the *P* and *M* genes, and the results showed that the recombinant virus SRV9 could be used as an effective expression vector for an exogenous gene. Jiao et al. [9] recombined feline herpesvirus (FHV-1) gB into a rabies virus SRV_9_ oral vaccine strain to construct a vaccine and rescued the expression of recombinant virus SRV_9_-FHV-gB. According to the findings, the vaccine was both safe and promoted the development of specific antibodies against FHV-1 and RABV in the NIMl bodies.

Hydatid disease (HD), also referred to as *Echinococcosis granulosa* (Eg), is a zoonotic parasitic disease that is caused by the infection of humans and animals by middle-stage larvae of *Echinococcus*, a parasite with a global distribution [10]. The WHO has classified it as a legally reported Class B animal disease [11]. Canines transmit cystic echinococcosis (CE) as the definitive host, and mammals such as humans, cows, and sheep as intermediate hosts are infected by contact with and ingesting food and water contaminated with Echinococcus eggs [12]. Canines serve as the sole definitive host of *Echinococcus granulosus*, the primary source of HD transmission. Furthermore, their excretory Echinococcus ova are the primary source of environmental contamination and infection in humans and other animals [13]. Therefore, it is necessary to regulate the infection of canines to reduce the transmission of HD.

Zhang et al. [14] used DDRT-PCR to identify specific gene families (EgM family genes: *EgM4*, *EgM9*, and *EgM123*) related to the development and number of bodies and eggs in *Echinococcus granulosus* at different developmental stages and expressed them through a prokaryotic expression system. Their antibody levels were also verified. Mao et al. [15]. analyzed the mRNA expression of EgM family genes at different developmental stages and found that the expression of the *EgM123* gene was significantly higher than that of the *EgM4* and *EgM9* genes at the same developmental stage, suggesting that the *EgM123* gene was the main gene carried by *Echinococcus granulosus* at the adult stage. Zhao et al. [16] immunized canines with EgM9 and EgM123 recombinant proteins in three batches, and specific IgG antibodies were detected in canine mesenteric lymph nodes and the small intestine, which triggered the production of anti-*Echinococcus granulosus* antibodies and increased the levels of IFN-γ and IL-10 in the canine serum to stimulate cellular and humoral immune responses. He et al. [17] recombinantly inserted the *Echinococcus granulosus EgM123* gene into the *Mycolicibacterium smegmatis* shuttle plasmid, transformed the recombinant plasmid into *Mycolicibacterium smegmatis*, and prepared a recombinant vaccine for the immunization of mice. After vaccination, specific antibodies in mouse serum continued to increase, which were significantly higher than those of the subunit vaccine control group after 8 weeks of immunization. Zhang et al. [14] emulsified the adult-specific proteins *EgM9* and *EgM123* of *Echinococcus granulosus* with Freund’s adjuvant, and IgG subclasses of IgG1 and IgG2 antibodies were detected in the serum of the immunized canine 45 d after infection. The populations of worms and egg production in the canine were markedly reduced after the canine’s euthanization.

Four auxiliary plasmids, including pcDNA4-N, pcDNA4-P, pcDNA4-G, pcDNA-L, and the pcDNA4-NPM+G_ΔCD_+EgM123+eGFP+L full-length plasmid, were established. Both virulent genes and the pseudogene ψ region in the CD region of the rabies virus SRV_9_ oral vaccine strain G were replaced with the *Echinococcus granulosus EgM123* gene by reverse genetics approaches, rescuing the “*EgM123* recombinant gene Rabies virus”. The findings of this study have the potential to decrease the infection rate of *Echinococcus granulosus* in the definitive host (canines) and to reduce the development and excretion of *Echinococcus granulosus* in canines. Thus, a comprehensive prevention and control system could be established from the source of infection to the susceptible population. Moreover, this recombinant virus can be used for the RV infection of canines, thereby realizing the purpose of “one vaccine against two diseases” and providing a comprehensive research idea for the research and application of “Rabies-HD bivalent recombinant gene vaccine”.

## 2. Materials and Methods

### 2.1. Materials

#### 2.1.1. Virus, Cells, and Serum

Rabies virus SRV_9_ was provided by the Academy of Military Medical Sciences Institute of Military Veterinary Medicine Ho Rongliang Research Fellows’ Gift; BSR cells were preserved in our laboratory and passed from 12 to 23 generations; BI fetal bovine serum was procured from Shanghai XP Biomed Ltd.

#### 2.1.2. Reagents

T4 DNA Ligase was purchased from Takara Biotechnology (Beijing) Co., Ltd., Beijing, China; *E. coli* DH5α competent cell was purchased from Nanjing Vazyme Biotechnology Co., Ltd., Nanjing, China; endotoxin-free plasmid extraction kit was purchased from Tiangen Biochemical Technology (Beijing) Co., Ltd., Beijing, China; KOD FX was purchased from East Yangfang (Shanghai) Biology Science and Technology Co., Ltd., Shanghai, China; cell culture medium (DMEM) and dual antibody were purchased from Wuhan Procell Technology Co., Ltd., Wuhan, China; Lipo6000TM transfection reagent was purchased from Shanghai Beyotime Biotechnology Co., Ltd., Shanghai, China; rabbit-derived rabies virus N protein muti-antibody was purchased from Biorby, UK; FITC-labeled goat anti-mouse and goat anti-rabbit was purchased from Beijing Bioss Co., Ltd., Beijing, China.

#### 2.1.3. Primer Design and Synthesis

The whole genome sequence of rabies virus SRV_9_ (GenBank No.: AF499686) was used to design the required primer (Table 1) using DNAMAN and primer 5, which was synthesized by Ykang Biotechnology Co., Ltd. (Urumqi, China).

### 2.2. Method

#### 2.2.1. Establishment of pcDNA4-N, P, G, L Auxiliary Plasmid of Rabies Virus

The *Knp* I/*Eco*R V restriction endonuclease site was introduced at both ends of the synthesized N (1353 bp), *P* (894 bp), and *G* (1575 bp) gene sequences, and the *Not* I/*Bstb* I restriction endonuclease site was introduced at both ends of the *L* (6384 bp) gene sequence. The synthesized *N*, *P*, *G*, and *L* genes were constructed into pcDNA4/myc-His vectors by T4 ligase and named pcDNA4-N, pcDNA4-P, pcDNA4-G, and pcDNA4-L, respectively. The ligated products were transformed into DH5α competent cells, and each auxiliary plasmid was extracted. The reconstructed recombinant plasmid was identified by PCR and the *Knp* I/*Eco*R V, *Not* I/*Bstb* I double-enzyme digestion method.

#### 2.2.2. Establishment of Eukaryotic Expression Vector of NPM Gene of Rabies Virus SRV_9_

The NPM (2967 bp) gene fragment was synthesized; the Kpn I site was introduced at the 5′ end of the gene, together with a 57 bp hammerhead-like nuclease HamRz sequence; and the EcoR V site was introduced at the 3′ end of the gene. The protein Linker-(GGGGS)2 was used to replace the spacer sequence between genes to ensure the normal expression of structural genes. The NPM recombinant gene was transferred to the pcDNA4/myc-HisB vector by T4 ligase and named pcDNA4-NPM. The ligated product was transformed into DH5α competent cells, and the plasmid was extracted and identified by PCR, plasmid double-enzyme digestion, and sequencing.

#### 2.2.3. Establishment of Eukaryotic Expression Vector of G_ΔCD_+EgM123+eGFP of Rabies Virus SRV_9_

In the synthesis of the long fragment G_ΔCD_+EgM123+eGFP, pseudogene ψ (132 bp) in the CD region of the *G* gene was deleted, and the *Echinococcus granulosus EgM123* gene (594 bp) was inserted. The green fluorescent protein eGFP gene (798 bp) was added, the *Eco*RV cleavage site was introduced at the 5′ end, the *Not* I cleavage site was introduced at the 3′ end, and protein Linker-(GGGGS) 2 was introduced between the gene sequences. The synthesized fragment G_ΔCD_+EgM123+eGFP was 2892 bp in size. The fragment was recombined into the pcDNA4/myc-HisB vector by T4 ligase and named pcDNA4-G_ΔCD_+EgM123+eGFP. The ligated product was transformed into DH5α competent cells, and the plasmid was extracted and identified by PCR, plasmid double-enzyme digestion, and sequencing.

#### 2.2.4. Establishment of N+P+M+G_ΔCD_+EgM123+eGFP Large Segment Vector of Rabies Virus SRV_9_

The pcDNA4-NPM recombinant plasmid was linearized by double-enzyme digestion using *Not I* and *EcoR* V restriction endonucleases. Both *Not I* and *EcoR V* restriction endonucleases were used to undergo double-enzyme digestion of pcDNA4-G_ΔCD_+EgM123+eGFP recombinant plasmids. The G_ΔCD_+EgM123+eGFP gene fragment was ligated into the pcDNA4-NPM vector using T4 ligase at 22 °C for 30 min and named pcDNA4-NPM+G_ΔCD_+EgM123+eGFP. The ligated product was transformed into DH5α competent cells, and the plasmid was extracted and identified by PCR, plasmid double-enzyme digestion, and sequencing.

#### 2.2.5. Establishment of NPM+G_ΔCD_+EgM123+eGFP+L Full-Length cDNA of Rabies Virus SRV_9_

The PCR method was used to amplify the pcDNA4-NPM+G_ΔCD_+EgM123+eGFP recombinant plasmid, and the size of the amplified NPM+G_ΔCD_+EgM123+eGFP target fragment was 5926 bp. After PCR was identified correctly, the PCR product was recovered by nucleic acid purification, and double-enzyme digestion of the pcDNA4-L recombinant plasmid was achieved by *Eco*R V and *Pme* I restriction endonucleases to obtain the pcDNA4-L linearized vector. The NPM+G_ΔCD_+EgM123+eGFP fragment was ligated with the pcDNA4-L linearized vector using the Gibson homologous recombination method at 50 °C for 60 min. Stbl3 competent cells were transformed with 10 μL of the ligated product. The pcDNA4-NPM+G_ΔCD_+EgM123+eGFP+L recombinant plasmid was extracted and identified through PCR, plasmid double-enzyme digestion, and sequencing.

#### 2.2.6. Rescue of Echinococcus Granulosus EgM123 Recombinant Gene Rabies Virus SRV_9_

BSR cells were passaged 24 h before electro-transformation, with a fusion degree of 70–80%. The cells were digested and re-suspended in a serum-free DMEM medium. After centrifugation at 1000 rpm for 5 min, the supernatant was discarded, and the cells were counted to 2 × 10^6^ using a cell counter plate, and 500 μL of electro-transformation buffer was added to re-suspend the cells. According to the plasmid concentration in Table 2, add the full-length cDNA plasmid and *N*, *P*, *G*, and *L* auxiliary plasmids in sequence to the BSR re-suspended cells, and then slowly add them into a 0.4 cm electro-transfer cup. Simultaneously, the full-length cDNA plasmid was constructed to control the negatively transfected cells, and electro-transformation was conducted (130 v, 25 ms, twice, 0.4 mm). The electro-transfer cup was placed in the incubator after cell electro-transformation and allowed to stand for 10 min. The liquid in the cup was gradually added to a 6-well plate, and the cells were incubated in serum-free DMEM for 6 h. After 6 h, the medium was replaced with a DMEM medium containing 10% fetal bovine serum and incubated for 72 h. The fluorescence signal changes were observed after 48 h of cultivation.

#### 2.2.7. Genetic Stability of Rescued Virus

The total RNA of the recombinant virus was extracted using the Trizol reagent, and the viral RNA was reverse-transcribed according to the Takara reverse transcription kit using the SRV_9_-N (F) gene-specific primer. The reaction mixture for reverse transcription comprised 1 μL of dNTPS, 2 μL of Mgcl_2_, 1 μL of 10X RT-buffer, 0.5 μL of AMV, 0.5 μL of SRV9-N(F), 0.25 μL of RNase Inhibitor, and RNA template (≤500 ng), as well as 10 μL of enzyme-free water. The reaction conditions were as follows: 42 °C, 30 min; 95 °C, 5 min; 5 °C, 5 min. Reverse transcription cDNA was used as the template and identified the structural genes of the recombinant virus using *N*, *P*, *M*, *G*, *EgM123*, and *eGFP* upstream and downstream primers, respectively. The PCR reaction mixture comprised 2.5 μL of 5X PCR Buffer, 0.25 μL of EX Taq, 0.25 μL of upstream primer, 0.25 μL of downstream primer, μL of cDNA2, and 7.5 μL of ddH_2_O. The reaction conditions were as follows: 95 °C, 5 min; 94 °C, 30 s; 58 °C, 30 s; 72 °C, 1 min/kb; 72 °C, 10 min; 4 °C insulation. Moreover, recombinant virus full-length cDNA plasmid and uninfected BSR cells were used as positive and negative controls, respectively. The size of the targeted strips was detected by nucleic acid gel electrophoresis.

#### 2.2.8. Morphological Characteristics of Recombinant Virus

BSR cells infected with the recombinant virus were scraped, centrifuged, double-fixed with 2.5% glutaraldehyde and 1% osmium acid, and dehydrated in a series of gradients of 30%, 50%, 70%, 80%, 90%, and 100% ethanol. The intracellular ethanol was then replaced by acetone. Tissue samples were immersed with 812 embedding agents and polymerized at 37 °C for 24 h, 45 °C for 24 h, and 60 °C for 24 h. Tissue was sectioned into ultrathin sections of 70 nm, double-stained with lead citrate and uranyl acetate, and intracellular viral morphology was examined using TEM.

#### 2.2.9. Titer of Recombinant Virus Measured by qRT-PCR

The viral fluids from all generations (9th, 10th, 11th, 12th, and 13th) were collected, and sample primers, EgM123 (F/R) along with the internal reference primer, GAPDH (F/R), were designed for qRT-PCR detection. Total viral RNA was extracted for reverse transcription; the qRT-PCR reaction system was configured as follows: 10 μL of 2X PerfectStartTM Green qPCR SuperMix (+Dye I/+Dye II), 0.6 μL of upstream primer f4 (10 μM), 0.6 μL of downstream primer (10 μM), 2 μL of cDNA template, and 6.8 μL of enzyme-free water. The reaction conditions were 94 °C, 30 s; 94 °C, 5 s; 60 °C, 15 s (signal collection). They were conducted over a total of 40 cycles, with 3 replicates per group, and the data were calculated by SPSS 27.0.1.

#### 2.2.10. Measurement of Median Lethal Dose (LD_50_) of Recombinant Virus

The concentrated 12th-generation viral solution was diluted at a ratio of 10^0^~10^−2^, and the dilution factor was 10. Three-day-old suckling rats were administered 0.03 mL of the viral solution intracerebrally via a microsyringe in a gradient of 10^0^~10^−2^ dilution while saline served as a negative control, with 3 replicates per group. The mortality of suckling rats in each group was counted on 21 consecutive days, and the virulence of the recombinant virus was calculated according to the Spearman–Karber method.

All animals were housed in a room with temperature and humidity controlled within an appropriate range, a light/dark 12 h cycle, free access to rat food, and drinking purified water in a clean cage. All animal experiments were approved by the Animal Experimental Ethics Committee of Xinjiang Agricultural University (license number: 2022037) and conducted by the “Guidelines for the Care and Use of Research Animals” formulated by the university.

## 3. Results

### 3.1. Identification of Auxiliary Transfection Plasmids of Rabies Virus SRV_9_

The auxiliary transfection plasmids of rabies virus SRV_9_ were amplified by PCR, and pcDNA4-N (1353 bp)-, pcDNA4-P (894 bp)-, pcDNA4-G (1575 bp)-, and pcDNA4-L (6384 bp)-targeted strips were amplified (Figure 1). The sequencing results of the amplified PCR products showed 100% compliance. Double-enzyme digestion was performed on the extracted recombinant plasmids pcDNA4-N, P, and G using *Knp* I and *EcoR* V restriction endonucleases. The vector enzyme fragment size was 5036 bp, the N gene enzyme fragment size was 1356 bp, the P gene enzyme fragment size was 894 bp, and the G gene enzyme fragment size was 1575 bp. The pcDNA4-L recombinant plasmid was identified through double-enzyme digestion using the *Not*I/*Bst*I restriction endonuclease. The vector enzyme fragment size was 5036 bp, and the *L* gene enzyme fragment size was 6384 bp (Figure 2).

### 3.2. Establishment of pcDNA4-N+P+M Eukaryotic Expression Vector and Identification of Recombinant Plasmids

The pcDNA4-NPM recombinant plasmid was confirmed with PCR, yielding targeted fragments of approximately 2979 bp (Figure 3). The recombinant plasmid pcDNA4-NPM underwent double-enzyme digestion analysis using *KpnI/EcoRv* restriction endonucleases, resulting in an NPM fragment of approximately 2979 bp and a vector fragment of 5036 bp (Figure 4). The pcDNA4-NPM recombinant plasmid sequencing results showed a compliance rate of 100%.

### 3.3. Identification of pcDNA4-G_ΔCD_+EgM123+eGFP Recombinant Plasmid

The pcDNA4-G_ΔCD_+EgM123+eGFP was confirmed via PCR, yielding targeted strips of approximately 2906 bp (Figure 5). The recombinant plasmid pcDNA4-G_ΔCD_+EgM123+eGFP underwent double-enzyme digestion analysis using *EcoR V/Not I* restriction endonucleases, resulting in a G_ΔCD_+EgM123+eGFP fragment of about 2906 bp and a vector fragment of 5064 bp (Figure 6). The pcDNA4-G_ΔCD_+EgM123+eGFP recombinant plasmid sequencing results showed a compliance rate of 100%.

### 3.4. Identification of pcDNA4-NPM+G_ΔCD_+EgM123+eGFP Recombinant Plasmid

The pcDNA4-NPM+G_ΔCD_+EgM123+eGFP recombinant plasmid underwent single-enzyme digestion analysis using the *Not I* restriction endonuclease, resulting in a band of 10,868 bp in size (Figure 7). The sequencing results of the pcDNA4-NPM+G_ΔCD_+EgM123+eGFP recombinant plasmid demonstrated a compliance rate of 100%.

### 3.5. Establishment of pcDNA4-NPM+G_ΔCD_+EgM123+eGFP+L Full-Length cDNA

The extracted full-length cDNA of recombinant plasmid underwent single-enzyme digestion analysis using the *Not I* restriction endonuclease, resulting in a band of 17,767 bp in size (Figure 8). The sequencing results of the full-length cDNA of the recombinant plasmid showed a compliance rate of 100% (Figure 9).

### 3.6. Observation of the Rescue Results of the Recombinant Virus Using a Fluorescence Microscope

The full-length cDNA of *EgM123* recombinant gene SRV_9_ carrying the exogenous gene and auxiliary plasmids pcDNA4-N, pcDNA4-P, pcDNA4-G, and pcDNA4-L were co-transfected into BSR cells by electro-transferring. After 48 h of transfection, only the full-length plasmid cDNA group was transfected, and the virus rescue group showed green fluorescence, while the normal BSR control group showed no fluorescence (Figure 10).

### 3.7. Detection of Structural Genes of Recombinant Virus by RT-PCR

The blind transmission of the sixth-generation recombinant virus was identified by RT-PCR using *N* (F/R), *P* (F/R), G (F/R), *EgM123* (F/R), and *eGFP* (F/R), and bands with 1353 bp, 894 bp, 1575 bp, 591 bp, and 798 bp were obtained, respectively (Figure 11). The gene sequence alignment results showed a compliance rate of 100%, suggesting that the recombinant virus gene was structurally complete, the virus rescue was successful, and the exogenous gene was genetically stable.

### 3.8. Characterization of Recombinant Virus by TEM

TEM analysis revealed that the cytoplasmic aggregates showed a bullet-like morphology, with viral particle sizes measuring 349.27 nm in length and 148.47 nm in diameter (Figure 12). This indicates that the recombinant virus retained the shape and structure characteristic of the RV, confirming the successful rescue of the Echinococcus granulosus *EgM123* recombinant gene rabies virus SRV_9_.

### 3.9. Titer Measurement of the Virus by qRT-PCR

The supernatants of the 9th-, 10th-, 11th-, 12th-, and 13th-generation viruses were analyzed for virus identification using qRT-PCR. However, the 11th, 12th, and 13th generations showed a substantial increase in viral expression relative to that of the 9th and 10th generations (*p* < 0.05). However, no significant increase in viral expression was detected between the 9th and 10th generations (*p* > 0.05) (Figure 13).

### 3.10. Virulence Measurement of Recombinant Virus LD_50_

The mortality status of suckling rats was observed for 21 days and the LD_50_ titer of the recombinant virus was calculated according to the Spearman–Karber method (Table 3).

## 4. Discussion

Canids serve as co-infected hosts, pathogen carriers, and transmitters, playing a vital intermediary role in the transmission of rabies and HD from animals to humans [18].

Currently, the international control of HD is mainly based on the prevention of *Echinococcus granulosus* infection in intermediate hosts (humans and animals) or the prevention of definitive hosts (dogs) by cutting off the developmental stages of *Echinococcus granulosus* [19]. In areas with a high prevalence of HD, dogs are mainly treated with anti-hydatid drugs such as albendazole and praziquantel [20]. After discontinuation of these drugs or their effects, dogs may remain susceptible to infection upon re-exposure to animal offal harboring cysts of Echinococcus granulosus, and prolonged use of these treatments may result in the emergence of drug resistance in the parasites [21]. Therefore, the optimal and most efficacious method for prevention and control is to vaccinate dogs to sustain the presence of anti-Echinococcus granulosus antibodies in the host’s system over an extended period, thereby interrupting the transmission of canine HD. Currently, vaccination is the primary method of prevention and control for rabies [22]. However, the technical challenges associated with immunizing canines by injection are the main cause of the difficulty in controlling rabies and its gradual spread, as dogs are frequently free-ranging in agricultural and pastoral areas. The most effective method of rabies prevention and control is the use of oral vaccines, which can be administered through direct feeding or field release [23].

Reverse genetics is the process of constructing an infectious cDNA clone of a virus, co-transfecting the cell with the cDNA and auxiliary plasmids, and obtaining a biologically functional virus in vitro, which is also known as “virus rescue”. The rescued recombinant virus is inoculated in the organism to express protective antigens and induce the organism to produce corresponding antibodies, thus achieving the purpose of immunization [24]. The construction of novel recombinant gene live vaccines with the RV as the vector has become the focus of genetic engineering vaccine research with the development and application of reverse genetics [25]. The RV has a wide range of applications as a vaccine virus vector for transmission. The genome of the RV is a negative-stranded and non-segmented RNA encoding five structural proteins: nucleoprotein (N), phosphorylated protein (P), matrix protein (M), glycoprotein (G), and RNA-dependent RNA polymerase (L), of which the G protein is the only antigen that stimulates the body to produce neutralizing antibodies against the RV [26]. Therefore, the study of the RV genetic engineering vaccine is mainly focused on the G protein. The viral nucleocapsid ensures the stability and large capacity of the exogenous gene, preventing its deletion in homologous recombination when the exogenous gene is expressed in the RV genome. The genome structure is simple, and the exogenous gene can be easily operated [27]. Replication of the RV is carried out in the cytoplasm of the cell and does not recombine with the chromosome of the host cell, which can induce humoral and cellular immunity, and has a strong protective effect on the organism. As a vector for genetic engineering vaccines, it has strong application advantages [28].

In this study, the SRV_9_ oral vaccine strain was used as a skeleton, the pseudogene in the CD region of the *G* gene was deleted, the *Echinococcus granulosus EgM123* gene was added, and the *Echinococcus granulosus EgM123* recombinant gene rabies virus SRV_9_ was rescued by reverse genetics approaches. Specifically, the NPM+G_ΔCD_+EgM123+eGFP gene was amplified by PCR, and the linearized vector was obtained by double-enzyme digestion of pcDNA4-L. The NPM+G_ΔCD_+EgM123+eGFP gene was constructed into the pcDNA4-L linearized vector by Gibson homologous recombination to achieve the seamless connection of *EgM123* recombinant gene rabies full-length cDNA. The efficacy of recombinant virus rescue was preliminarily confirmed by direct fluorescence observation after the full-length cDNA and auxiliary plasmids were co-transfected into BSR cells via electro-transformation. The recombinant virus’s genetic stability was confirmed by the amplification of each gene fragment for each gene fragment by RT-PCR after blind transmission for six generations. The virus particles were observed to be in the shape of bullets by TEM, indicating that the pseudogene region between *G* and *L* in the SRV_9_ full-length cDNA was replaced by an exogenous gene, which did not affect the virus’s morphology [29]. This morphological confirmation further confirms the success of the virus rescue in this study. The virus expression levels in the 9th, 10th, 11th, 12th, and 13th generations of the continuous cultures were quantified using qRT-PCR. The viral mRNA expression levels were recorded as 0.97, 1.09, 1.611, 1.287, and 1.699, respectively, and viral mRNA expression levels showed an increasing trend, suggesting that the *EgM123* recombinant gene RV was successfully rescued; the three-day-old suckling rat LD_50_ assay determined that the recombinant virus titer was 10^−1.4^/0.03 mL and the original virus titer was 10^−6.7^/0.03 mL [30], suggesting that the recombinant virus constructed by reverse genetics approaches had a weaker virulence compared with that of the parental SRV_9_ strain. In conclusion, this study effectively rescues the *EgM123* recombinant gene RV strain, thereby establishing the basis for future research on the rabies novel vaccine and *EgM123*.

## Figures and Tables

**Figure 1 viruses-17-00030-f001:**
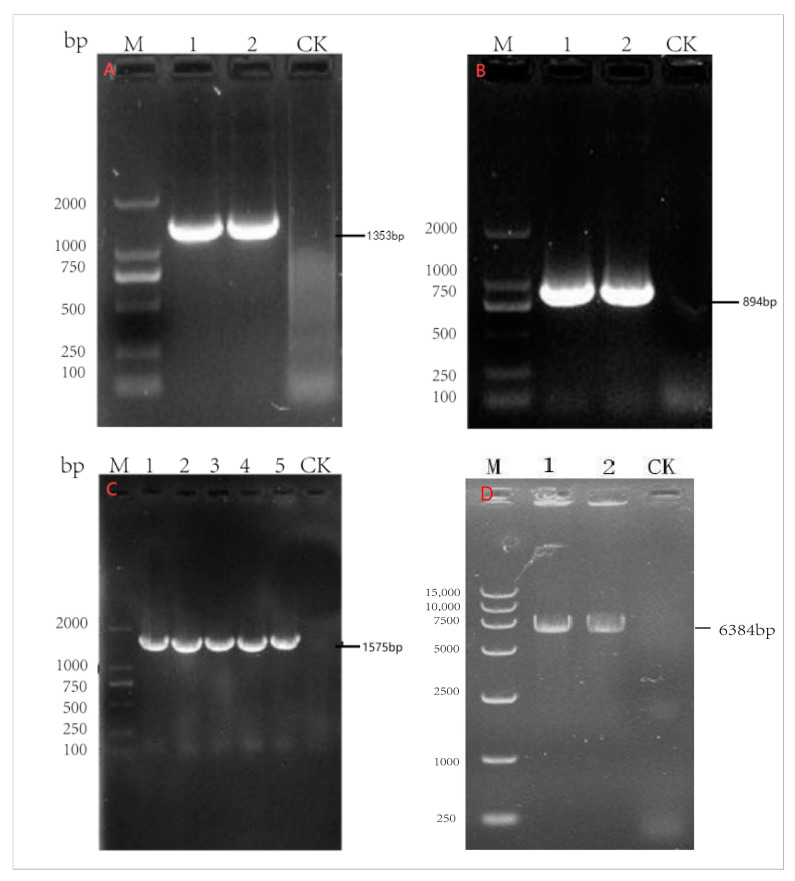
Identification of auxiliary plasmids of pcDNA4-N, pcDNA4-P, pcDNA4-G, pcDNA4-L recombinant genes by PCR. Note: (**A**) pcDNA4-P recombinant plasmid; M: 2000 DNA Marker; 1–2: PCR amplification of pcDNA4-N; CK: pcDNA4-N negative control. (**B**) pcDNA4-P recombinant plasmid; M: 2000 DNA Marker; 1–2: PCR amplification of pcDNA4-P; CK: pcDNA4-P negative control. (**C**) pcDNA4-G recombinant plasmid; M: 2000 DNA Marker; 1–5: PCR amplification of pcDNA4-G; CK: pcDNA4-G negative control. (**D**) pcDNA4-L recombinant plasmid; M: 15,000 DNA Marker; 1–2: PCR amplification of pcDNA4-L; CK: pcDNA4-L negative control.

**Figure 2 viruses-17-00030-f002:**
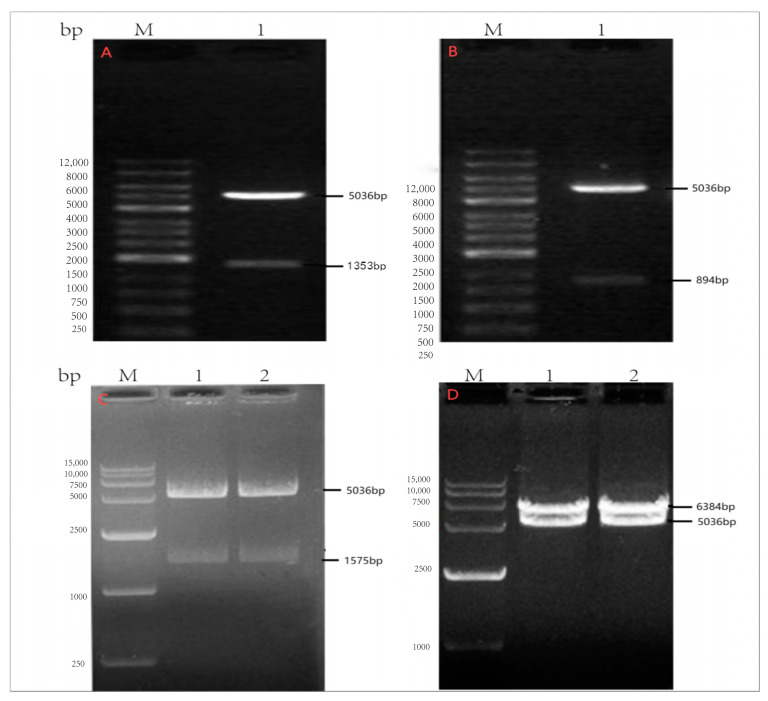
Double-enzyme digestion identification of pcDNA4/myc-N, pcDNA4/myc-P, pcDNA4/myc-G, pcDNA4/myc-L recombinant gene plasmid. Note: (**A**) pcDNA4/myc-N; M: 12,000 DNA Marker; 1: pcDNA4/myc-N enzyme digestion. (**B**) pcDNA4/myc-P; M: 12,000 DNA Marker; 1: pcDNA4/myc-P enzyme digestion. (**C**) pcDNA4/myc-G; M: 15,000 DNA Marker; 1–2: pcDNA4/myc-G enzyme digestion. (**D**) pcDNA4/myc-L; M: 15,000 DNA Marker; 1–2: pcDNA4/myc-L enzyme digestion.

**Figure 3 viruses-17-00030-f003:**
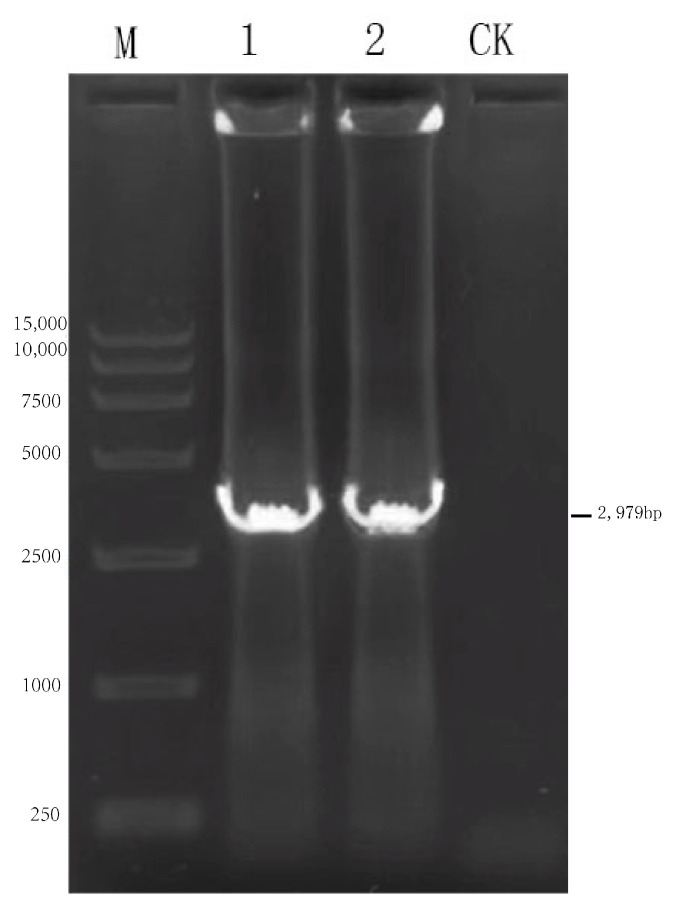
Identification of bacterial solution of rabies virus pcDNA4-NPM+G_ΔCD_+EgM123+eGFP. Note: M: 15,000 DNA Marker; 1–2: PCR amplification of pcDNA4- N+P+M; CK: pcDNA4-N+P+M negative control.

**Figure 4 viruses-17-00030-f004:**
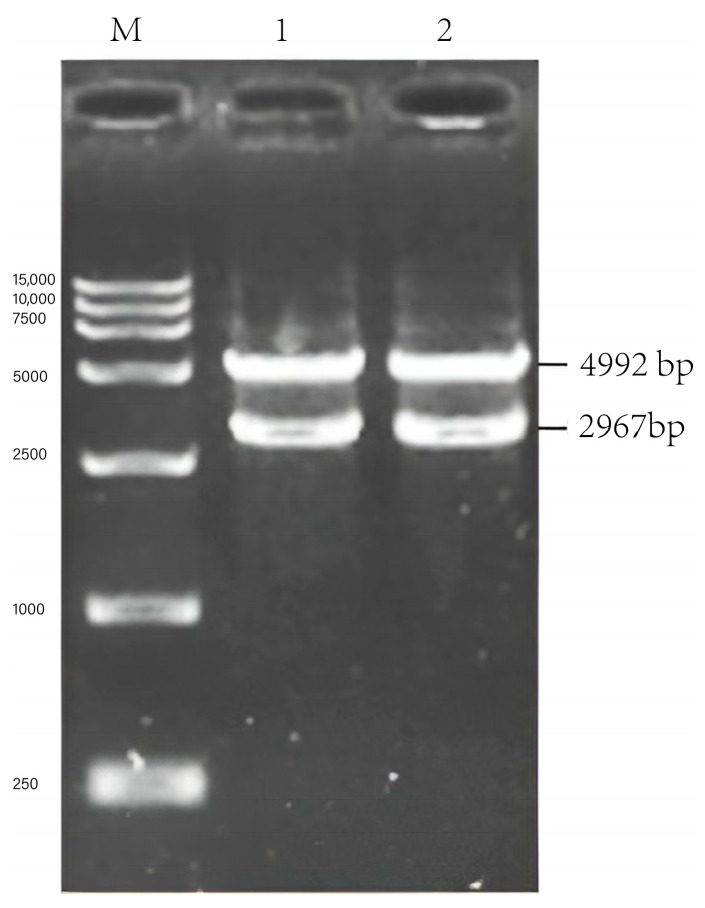
Identification of rabies virus pcDNA4-NPM+G_ΔCD_+EgM123+eGFP recombinant plasmid by enzymatic digestion. Note: M: 15,000 DNA Marker; 1–2: pcDNA4-NPM+G_ΔCD_+EgM123+eGFP plasmid enzymatic digestion results.

**Figure 5 viruses-17-00030-f005:**
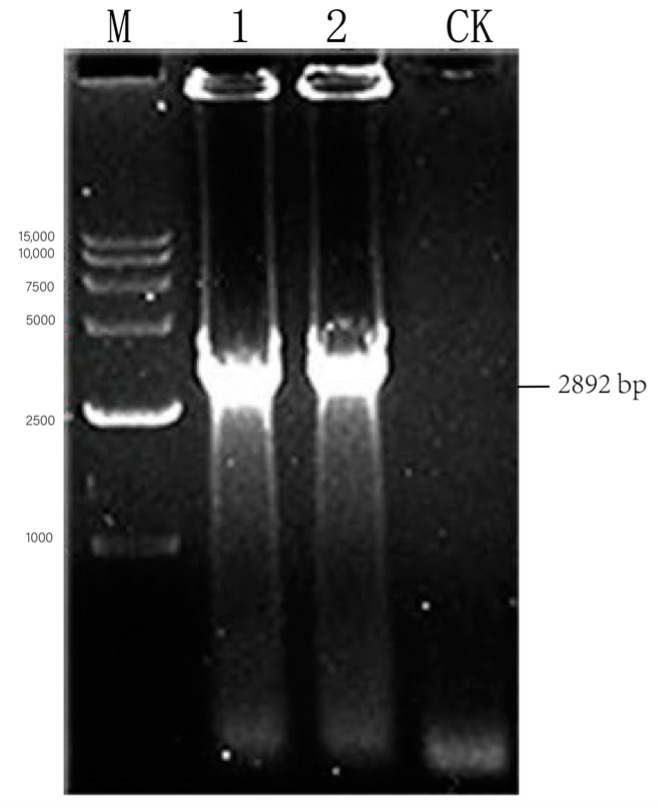
PCR identification of pcDNA4-G_ΔCD_ + EgM123+eGFP recombinant plasmid. Note: M: 15,000 DNA Marker; 1–2: PCR amplification of pcDNA4-G_ΔCD_+EgM123+eGFP; CK: pcDNA4-G_ΔCD_+EgM123+eGFP negative control.

**Figure 6 viruses-17-00030-f006:**
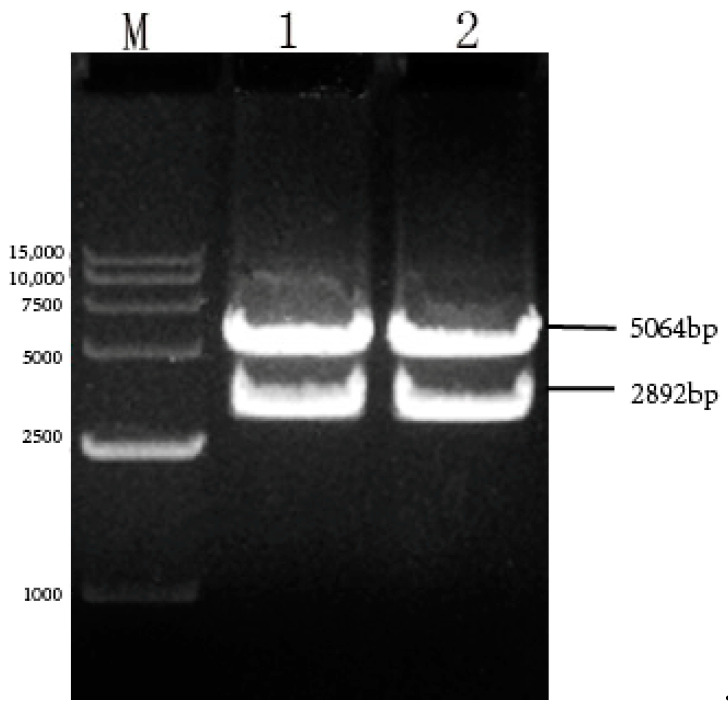
Identification of rabies virus pcDNA4-G_ΔCD_+EgM123+eGFP recombinant plasmid by enzymatic digestion. Note: M: 15,000 DNA Marker; 1–2: pcDNA4-G_ΔCD_+EgM123+eGFP plasmid enzymatic digestion results.

**Figure 7 viruses-17-00030-f007:**
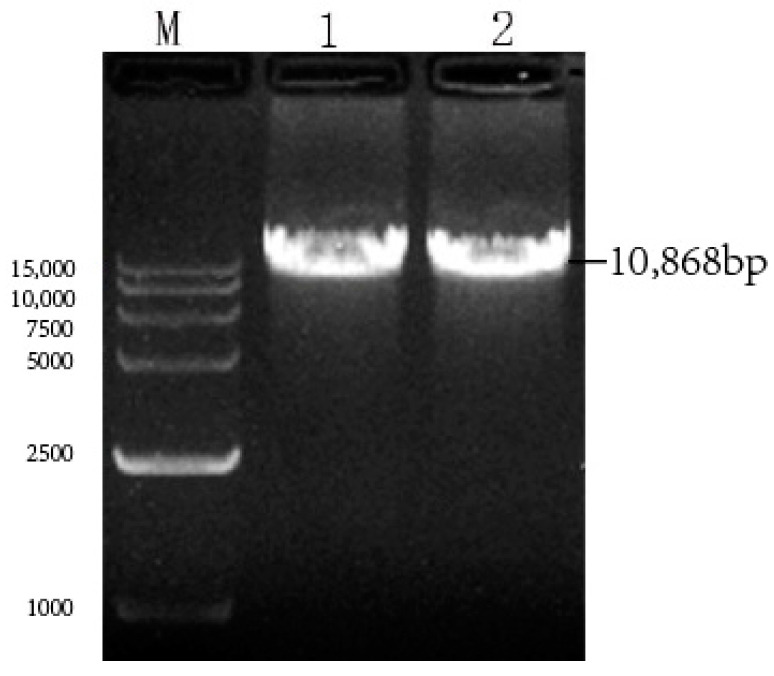
Identification of pcDNA4-NPM+G_ΔCD_+EgM123+eGFP recombinant plasmid by zymography. Note: M: 15,000 DNA Marker; 1–2: pcDNA4-NPM+G_ΔCD_+EgM123+eGFP plasmid enzymatic digestion results.

**Figure 8 viruses-17-00030-f008:**
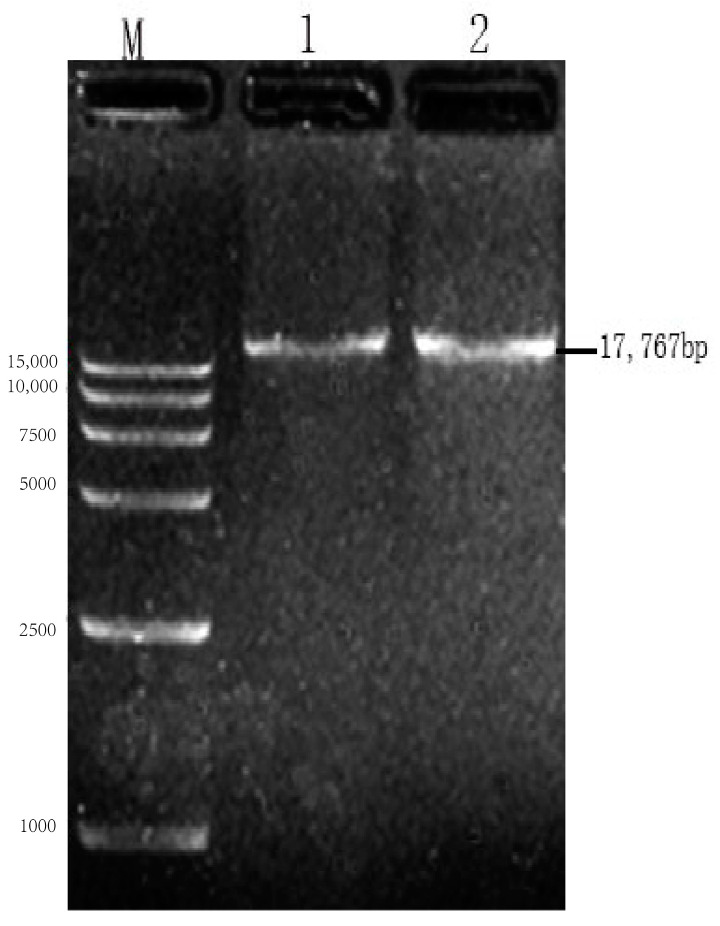
Identification of rabies virus pcDNA4-NPM+G_ΔCD_+EgM123+eGFP+L recombinant plasmid digestion. Note: M: 15,000 DNA Marker; 1–2: pcDNA4-NPM+G_ΔCD_+EgM123+eGFP+L plasmid enzymatic digestion results.

**Figure 9 viruses-17-00030-f009:**
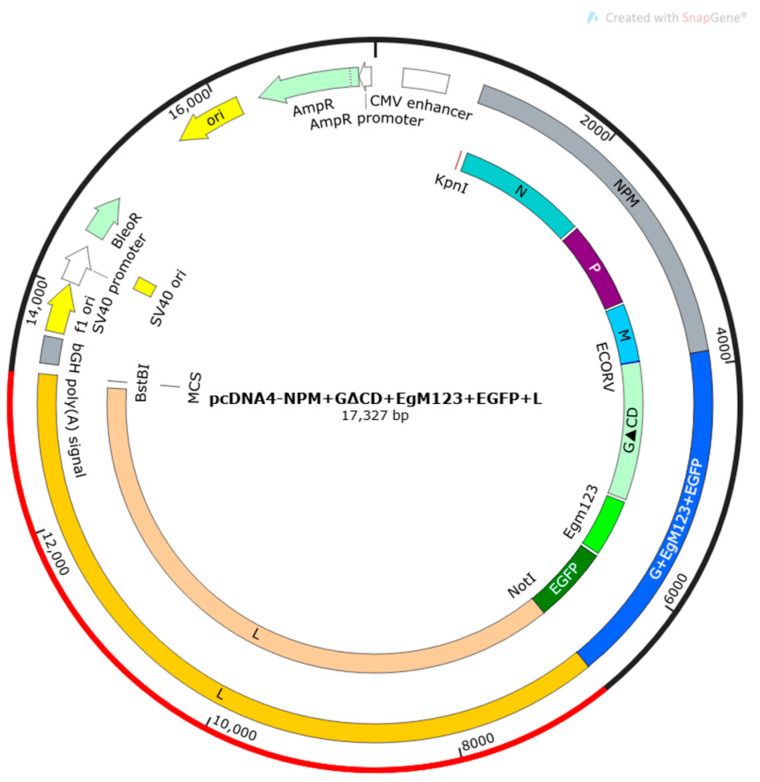
Plasmid mapping of rabies viruspcDNA4-NPM+G_ΔCD_+EgM123+eGFP+L.

**Figure 10 viruses-17-00030-f010:**
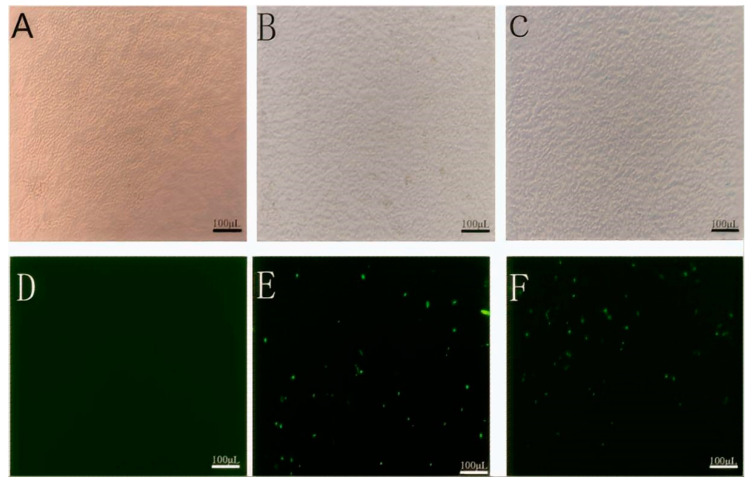
Observation of the rescue results using a fluorescence microscope.Note: (**A**): Normal BSR cells; (**B**): full-length cDNA transfection; (**C**): virus rescue; (**D**): observation of normal BSR cells using fluorescence microscope; (**E**): observation of transfected cDNA cells using fluorescence microscope; (**F**): observation of virus rescue cells using fluorescence microscope.

**Figure 11 viruses-17-00030-f011:**
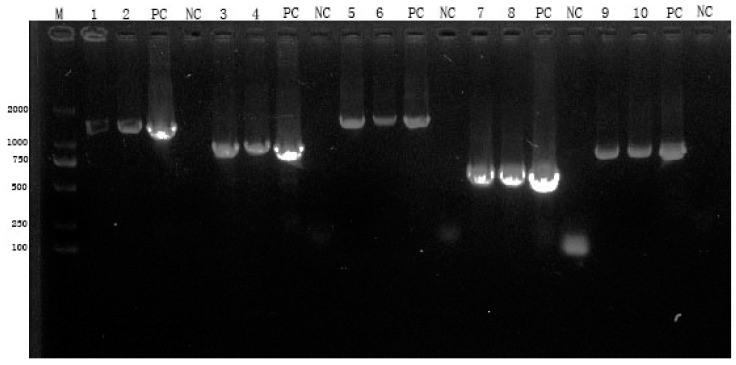
Virus identification by RT-PCR. Note: M: DL2000 bp marker; 1, 2: N-fragment virus identification; 3, 4: P-fragment virus identification; 5, 6: GΔCD-fragment virus identification; 7, 8: EgM123-fragment virus identification; 9, 10: eGFP-fragment virus identification; PC: corresponding positive plasmid control; NC: corresponding uninfected virus negative control.

**Figure 12 viruses-17-00030-f012:**
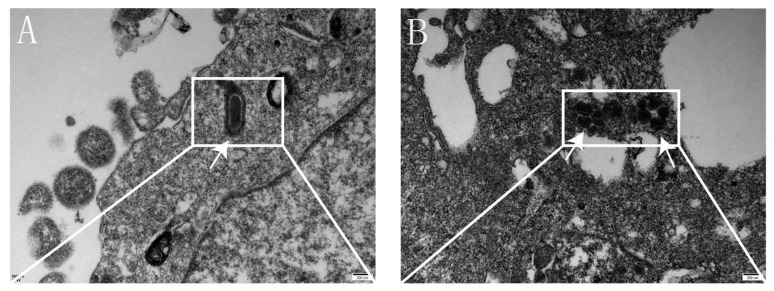
TEM images of recombinant virus. Note: (**A**): Slug shape of rabies virus; (**B**): Virus particles accumulate in the small pool of rough endoplasmic reticulum.

**Figure 13 viruses-17-00030-f013:**
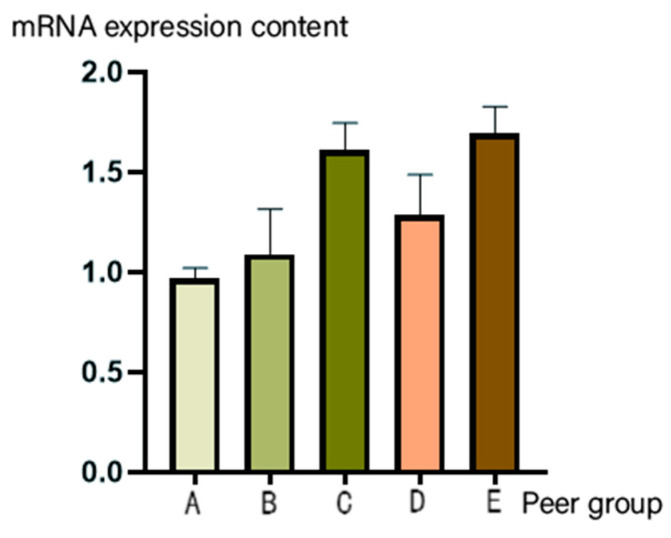
Virus identification by fluorescence quantitative PCR. Note: A: 9th-generation virus; B: 10th-generation virus; C: 11th-generation virus; D: 12th-generation virus; E: 13th-generation virus.

**Table 1 viruses-17-00030-t001:** Primers used in this study.

Primer Name	Nucleotide Sequence (5′-3′)
N (F)	GGTACCCTACAATGGATGCCGACAAG
N (R)	GATATCTCAACTTCTTATGAGTCACTCG
P (F)	GGTACCCATGAGCAGATCTTTGTCAAT
P (R)	GATATCTCGGTTAGCAAGATGTATAGCGATT
G (F)	GGTACCAGGAAAGATGGTTCCTCAGGCTCTC
G (R)	GATATCTTACAGTCTGGTCTCACCCCAC
L (F)	GCGGCCGCATGCTCGATC
L (R)	GTGAGCCTACCGATAAGCTT
NPM (F)	GGTACCTGTTAAGCGTCTGATG
NPM (R)	GATATCTTATTCTAGAAGCAG
EGM123 (F)	GTGAATTTTGCCTGCCCGTT
EGM123 (R)	AGCACAACCTCAGTCATGGG

**Table 2 viruses-17-00030-t002:** Amount of electroporation plasmid.

Plasmid Name	Initial Concentration (ng/μL)	Amount Used (μL)	Mass (μL)
pcDNA4-N	1.43126	1	1.5
pcDNA4-P	1.41548	1.1	1.5
pcDNA4-G	1.39627	0.7	1
pcDNAg-L	1.21937	0.82	1
pcDNA4-NPM+G_ΔCD_+EgM123+eGFP+L (full-length cDNA)	0.94863	4.22	4

**Table 3 viruses-17-00030-t003:** Virulence measurement of recombinant virus LD_50._

Group	Virus Dilution Ratio	Number of Suckling Rats	Dose	Number of Deaths (Rats)	Mortality Rate *p* (%)
12th generation	10^0^	14	0.03 mL/rat	13	92.86%
	10^−1^	18	0.03 mL/rat	11	61.11%
	10^−2^	9	0.03 mL/rat	3	33.33%
LD50 titer	10^−1.4^/0.03 mL				

## Data Availability

The datasets supporting the findings of this article are included within the paper and its supplementary materials.

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
