# Peer review of "Establishment of Echinococcus granulosus EgM123 Recombinant Gene Rabies Virus SRV9 and Identification of Its Biological Characteristics"

_viruses, 2024, doi:10.3390/v17010030_

Round 1
Reviewer 1 Report
Comments and Suggestions for Authors
The authors present a preliminary report on the development of a new bivalent rabies vaccine targeting rabies and hydatid disease. The authors demonstrate the development of the vaccine candidate and preliminary safety data. The experiments are well described and this is a promising platform to combat two severe diseases.
Considerations for modifications:
Section 2.1.1 provide original source and passage range of BSR and SRV9
Section 2.2.10 line 231 and again 233, clarify units and dilution scheme. The Table suggests a serial 10fold dilution the use of ~ suggests "roughly" which is imprecise. Imprecision will likely lead to difficulties in repeating the experiments for further development of the vaccine. If GFP is present and rabies G is present an FFU can be used to establish titer.
Why is GFP retained? Serves no purpose and takes up space that may contribute to instability of the genome.
For the LD50 comparison to the SRV9 parental is preferred to establish a safety potential comparative.
The candidacy of the vaccine would be significantly strengthened by immunogenicity data, which could be accomplished with the LD50 mice samples. Recommend inclusion of at least antibody titer, if not also t-cell data.
An animal care and use committee clearance statement must be included for ethical reasons.
EM is described but is not shown, please show the EM in the main paper.
Comments on the Quality of English Language
The manuscript contains several statements that suggest the use of AI to assist writing. If AI was used, please provide the methodology and ensure that it falls within the journals guidelines for use of AI.
Line 37. Aquaphobia is a generalized fear of water. Hydrophobia is often used to describe rabies.
Line 37 neuroleptic connotes responses to overstimulation based on increased or decreased neurotransmitters frequently associated with over use or improper psychiatric treatment and is associated with other conditions. Consider rephrasing to say furious or paralytic form which is common language for rabies presentation.
Line 76, remove "immune" from "immune antibodies" or clarify what is meant, neutralizing, etc.?
Consistency of SRV9 or SRV9 with the 9 subscripted.
Author Response
On behalf of all co-authors, I would like to thank reviewers very much for favorable comments and constructive suggestions on our manuscript (MS) ID viruses-3340331. The reviewers considered our MS to be of general interest to the readership of viruses, and recommended the acceptance of our MS for publication after revisions.
We have revised our MS strictly according to the reviewers’ comments and suggestions. We used the “tracked changes” mode in the WORD to show the revised/changed text in the revised MS. Two MS files are uploaded: the “clean version” and the one showing “tracked changes”. In the following, we detail our point-by-point responses to the reviewers’ comments and suggestions. We made all our responses in blue colour for clarity.
Responses to comments and suggestions to Reviewer #1
General comments:
Section 2.1.1 provide original source and passage range of BSR and SRV9
Response:Agreed and revised accordingly.
General comments:
Section 2.2.10 line 231 and again 233, clarify units and dilution scheme. The Table suggests a serial 10fold dilution the use of ~ suggests "roughly" which is imprecise. Imprecision will likely lead to difficulties in repeating the experiments for further development of the vaccine. If GFP is present and rabies G is present an FFU can be used to establish titer.
Response:We are very sorry for this mistake and we have corrected it accordingly.
General comments:
Why is GFP retained? Serves no purpose and takes up space that may contribute to instability of the genome.
Response:In order to facilitate the observation of the rescue of the recombinant virus, the enhanced green fluorescent protein gene sequence was added after the EgM123 gene of Echinococcus granulosus.
General comments:
For the LD50 comparison to the SRV9 parental is preferred to establish a safety potential comparative.
Response:We are very grateful to the No.1 reviewer for his valuable and constructive suggestions on our manuscript. Suckling mice ( including 3-day-old suckling mice ) are susceptible to the rabies virus, so they are used as experimental animals to determine the LD50 virus titer.
General comments:
The candidacy of the vaccine would be significantly strengthened by immunogenicity data, which could be accomplished with the LD50 mice samples. Recommend inclusion of at least antibody titer, if not also t-cell data.
Response:We sincerely thank the No.1 reviewer for their valuable and constructive suggestions on our manuscripts, and we fully agree with your views. The core of this paper focuses on constructing a recombinant virus and identifying its biological characteristics. In the following research stage, we will carry out in-depth research on the immunization of recombinant virus vaccines, aiming to obtain the immunogenicity data of recombinant viruses more systematically, thereby further enhancing their potential as vaccine candidates.
General comments:
An animal care and use committee clearance statement must be included for ethical reasons.
Response:We are very sorry for this mistake and attach an application for ethical review of laboratory animal welfare to the annex.
General comments:
EM is described but is not shown, please show the EM in the main paper.
Response:We are very sorry for this error and attach the modified EM to the attachment.
General comments:
Line 37: Aquaphobia is a generalized fear of water. Hydrophobia is often used to describe rabies.
Response:We are very sorry for this mistake and we have corrected it accordingly.
General comments:
Line 37 neuroleptic connotes responses to overstimulation based on increased or decreased neurotransmitters frequently associated with over use or improper psychiatric treatment and is associated with other conditions. Consider rephrasing to say furious or paralytic form which is common language for rabies presentation.
Response:Agreed and revised accordingly.
General comments:
Line 76, remove "immune" from "immune antibodies" or clarify what is meant, neutralizing, etc.?
Response:We are very sorry for this mistake and we have corrected it accordingly.
General comments:
Consistency of SRV9 or SRV9 with the 9 subscripted.
Response:We are very sorry for this mistake and we have corrected it accordingly.
Reviewer 2 Report
Comments and Suggestions for Authors
This is an interesting article reporting on the establishment of Echinococcus granulosus EgM123 recombinant 1 gene Rabies virus SRV9 and identification of its biological characteristics. Based on the economic losses in husbandry resulting from rabies and HD, an effective and efficient bivalent oral vaccine for preventing HD and rabies is urgently required. The synthesized EgM123 recombinant gene Rabies virus SRV9 can function as a vaccine strain for the development of the "Rabies-HD bivalent recombinant gene oral vaccine," contributing to the prevention and management of rabies and HD in animals.
Required revisions are as follows:
Line 37: Please use hydrophobia instead of aquaphobia
Line 39-42: one sentence is enough. This is a repetition.
Line 62-71: please revise this section properly distinguishing Echinococcus granulosus from Echinococcus multilocularis and related diseases, i.e., Hydatid Disease and Alveolar Echinococcosis respectively.
Line 64: I suppose this is Echinococcus granulosus not Echinococcus multilocularis
Line 101: secretion, consider replacement by excretion
Line 405-406: These are the names of the authors of this reference:
Kumar A, Bhatt S, Kumar A, et al. Canine rabies: An epidemiological significance, pathogenesis, diagnosis, prevention, and 405 public health issues. Comparative immunology, microbiology and infectious diseases, 2023, 97101992-101992.
Line 415-416: These are the names of the authors of this reference:
Yan W, Xiang R, Chen J, et al. Proteomics analysis of suckling mouse brain infected with attenuated rabies virus strain SRV9. Acta 415 Virologica, 2019, 63(4): 423-432.
Line 423-424: Please cite a more recent document, this one is dated 2014
Line 425: the first author should be cited as: Alvarez Rojas C A, ….
Line 427-428: These are the names of the authors of this reference:
Porcu F, Cantacessi C, Dessì G, et al. 'Fight the parasite': raising awareness of cystic echinococcosis in primary school 427 children in endemic countries. Parasites & Vectors, 2022, 15(1):449-449.
Line 449: This is how to cite the authors of this reference:
Jabbar A, Ahmadi A, Irm N, et al. Rabies in Pakistan: Policies and recommendations. Public Health Challenges, 2024, 3(1).
Line 454-455: These are the names of the authors of this reference:
Yankowski C, Wirblich C, Kurup D, et al. Inactivated rabies-vectored SARS-CoV-2 vaccine provides long-term immune 454 response unaffected by vector immunity. NPJ Vaccines, 2022, 7(1): 110-110.
Some editing of the text is also needed.
Author Response
On behalf of all co-authors, I would like to thank reviewers very much for favorable comments and constructive suggestions on our manuscript (MS) ID viruses-3340331. The reviewers considered our MS to be of general interest to the readership of viruses, and recommended the acceptance of our MS for publication after revisions.
We have revised our MS strictly according to the reviewers’ comments and suggestions. We used the “tracked changes” mode in the WORD to show the revised/changed text in the revised MS. Two MS files are uploaded: the “clean version” and the one showing “tracked changes”. In the following, we detail our point-by-point responses to the reviewers’ comments and suggestions. We made all our responses in blue colour for clarity.
Responses to comments and suggestions to Reviewer #2
General comments:
Line 37: Please use hydrophobia instead of aquaphobia.
Response:Agreed and revised accordingly.
General comments:
Line 39-42: one sentence is enough. This is a repetition.
Response:We are very sorry for this mistake and we have corrected it accordingly.
General comments:
Line 62-71: please revise this section properly distinguishing Echinococcus granulosus from Echinococcus multilocularis and related diseases, i.e., Hydatid Disease and Alveolar Echinococcosis respectively.
Response:We thank the reviewer very much for constructive suggestions and we have distinguish between Echinococcus granulosus and Echinococcus multilocularis and related diseases.
General comments:
Line 64: I suppose this is Echinococcus granulosus not Echinococcus multilocularis.
Response:We are very sorry for this mistake and we have corrected it accordingly.
General comments:
Line 101: secretion, consider replacement by excretion.
Response: Agreed and revised accordingly.
General comments:
Line 405-406: These are the names of the authors of this reference:
Kumar A, Bhatt S, Kumar A, et al. Canine rabies: An epidemiological significance, pathogenesis, diagnosis, prevention, and 405 public health issues. Comparative immunology, microbiology and infectious diseases, 2023, 97101992-101992.
Response:We are very sorry for this mistake and we have corrected it accordingly.
General comments:
Line 415-416: These are the names of the authors of this reference:
Yan W, Xiang R, Chen J, et al. Proteomics analysis of suckling mouse brain infected with attenuated rabies virus strain SRV9. Acta 415 Virologica, 2019, 63(4): 423-432.
Response:We are very sorry for this mistake and we have corrected it accordingly.
General comments:
Line 423-424: Please cite a more recent document, this one is dated 2014.
Response:We have updated the article into 3 years and will replace it.
General comments:
Line 425: the first author should be cited as: Alvarez Rojas C A, ….
Response:We are very sorry for this mistake and we have corrected it accordingly.
General comments:
Line 427-428: These are the names of the authors of this reference:
Porcu F, Cantacessi C, Dessì G, et al. 'Fight the parasite': raising awareness of cystic echinococcosis in primary school 427 children in endemic countries. Parasites & Vectors, 2022, 15(1):449-449.
Response:We are very sorry for this mistake and we have corrected it accordingly.
General comments:
Line 449: This is how to cite the authors of this reference:
Jabbar A, Ahmadi A, Irm N, et al. Rabies in Pakistan: Policies and recommendations. Public Health Challenges, 2024, 3(1).
Response:We are very sorry for this mistake and we have corrected it accordingly.
General comments:
Line 454-455: These are the names of the authors of this reference:
Yankowski C, Wirblich C, Kurup D, et al. Inactivated rabies-vectored SARS-CoV-2 vaccine provides long-term immune 454 response unaffected by vector immunity. NPJ Vaccines, 2022, 7(1): 110-110.
Response:We are very sorry for this mistake and we have corrected it accordingly.
Round 2
Reviewer 1 Report
Comments and Suggestions for Authors
Authors have addressed all prior concerns adequately.
Please double check references, for Ref 26, Given names are listed first rather than family/surname.
Please add statement to 2.2.10 that all in-vivo experiments were approved by the Institutional Animal Care and use Committee or equivalent oversight body.
Author Response
Re: Revised Manuscript ID viruses-3340331
On behalf of all co-authors, I would like to thank you and the two reviewers very much for favorable comments and constructive suggestions on our manuscript (MS) ID viruses-3340331. The reviewers considered our MS to be of general interest to the readership of viruses, and recommended the acceptance of our MS for publication after revisions.
We have revised our MS strictly according to the reviewers’ comments and suggestions. We used the “tracked changes” mode in the WORD to show the revised/changed text in the revised MS. Two MS files are uploaded: the “clean version” and the one showing “tracked changes”. In the following, we detail our point-by-point responses to the reviewers’ comments and suggestions. We made all our responses in blue colour for clarity.
Responses to comments and suggestions to Reviewer #1
General comments:
Please double check references, for Ref 26, Given names are listed first rather than family/surname.
Response:We are very sorry for this mistake and we have corrected it accordingly.
General comments:
Please add statement to 2.2.10 that all in-vivo experiments were approved by the Institutional Animal Care and use Committee or equivalent oversight body.
Response:Agreed and revised accordingly.